# Standardized evaluation of diabetic retinopathy using artificial intelligence and its association with metabolic dysfunction-associated steatotic liver disease in Japan: A cross-sectional study

Koji Komatsu[1]*, Kei Sano[1,2], Kota Fukai[2], Ryo Nakagawa[3], Takashi Nakagawa[3], Masayuki Tatemichi[2], Tadashi Nakano[1]

1 Department of Ophthalmology, Jikei University School of Medicine, Tokyo, Japan, 2 Department of Preventive Medicine, School of Medicine, Tokai University, Kanagawa, Japan, 3 Omiya City Clinic, Saitama, Japan

* koji.0406.kk0000@gmail.com

## Abstract

Metabolic dysfunction-associated steatotic liver disease (MASLD) is common in patients with obesity and diabetes and can lead to serious complications. This study aimed to evaluate fundus photographs using artificial intelligence to explore the relationships between diabetic retinopathy (DR), MASLD, and related factors. In this cross-sectional study, we included 1,736 patients with a history of diabetes treatment or glycated hemoglobin (HbA1c) levels of $\geq$6.5%. All participants were negative for hepatitis B surface antigen and hepatitis C virus antibody and were selected from 33,022 examinees at a health facility in Japan. Fundus photographs were analyzed using RetCAD software, and DR scores were quantified. The presence of DR was determined using two cutoffs: sensitivity (CO20) and specificity (CO50). Steatotic liver (SL) stages were assessed via ultrasound and fibrosis indices and classified into three groups: no SL (SL0), SL with low fibrosis (SL1), and SL with high fibrosis (SL2). Odds ratios (ORs) for the risk of DR associated with each SL stage were calculated using logistic regression, adjusted for age, sex, body mass index, HbA1c, C-reactive protein level, and alcohol consumption. The risk of DR was lower in the SL1 (OR: 0.63, 0.54) and SL2 (OR: 0.64, 0.77) groups compared to the SL0 group at CO20 for both the Fibrosis-4 Index (FIB-4) and the non-alcoholic fatty liver disease fibrosis score (NFS), respectively. Additionally, higher levels of cholinesterase were consistently associated with a reduced risk of DR (FIB-4: OR 0.52, NFS: OR 0.54) at CO50. This study demonstrates that MASLD was associated with a reduced risk of DR, with cholinesterase levels providing further protective effects, highlighting the need for further research into the protective mechanisms and refinement of DR evaluation techniques. The standardized AI evaluation method for DR offers a reliable approach for analyzing retinal changes.

**Data Availability Statement:** Data cannot be shared publicly because of ethical restrictions. Data are available from the Institutional Review Board of the Jikei University School of Medicine for researchers who meet the criteria for access to confidential data. The Jikei University Hospital Ethics Committee Secretariat E-mail: rinri@jikei.ac.jp 3-25-8 Nishi-Shimbashi, Minato-ku Tokyo, Japan, 105-8461 TEL:+81-3-3433-1111.

**Funding:** The author(s) received no specific funding for this work.

**Competing interests:** NO authors have competing interests.

## Introduction

Metabolic dysfunction-associated steatotic liver disease (MASLD) is primarily caused by life-style-related conditions such as obesity, diabetes, dyslipidemia, and genetic factors, rather than by liver damage resulting from excessive alcohol consumption, hepatitis viruses, or drug use [1]. MASLD affects 30% of the general adult population and 60–70% of patients with diabetes and obesity [2]. As MASLD progresses, it can lead to metabolic dysfunction-associated steato-hepatitis and cirrhosis, significantly increasing the risk of liver cancer [3]. Moreover, MASLD can increase the incidence of cardiovascular disease, a major macrovascular complication, in patients with type 2 diabetes [4].

Diabetic retinopathy (DR), a common complication of diabetes mellitus (DM), has frequently been reported in association with MASLD. In MASLD, DR has been identified as an independent risk factor for hepatocellular carcinoma, drawing significant attention to the relationship between the two conditions [5]. However, previous studies have shown inconsistent associations between DR and MASLD varies across previous reports [6]. A meta-analysis investigating the relationship between MASLD and DR reported no overall association, although a varying trend in different countries was observed [7].

Two major factors possibly contribute to the discrepancies in outcomes between DR and MASLD. First, inconsistencies in diagnoses could be a contributing factor. Identifying risk factors for DR in the general population is challenging, and the varying experience and skill levels of ophthalmologists in interpreting fundus images can lead to misclassifications. Kawasaki et al. conducted a study on 1,221 participants and reported a kappa value of 0.56, indicating moderate agreement between a local ophthalmologist and two retinal specialists [8]. Similarly, Hashimoto et al. conducted a study with included 1,806 patients, where diagnostic decisions were made by two ophthalmologists, and in cases of disagreement, three retinal specialists were consulted [9], further highlighting the inconsistency in outcomes. Second, the stage of liver fibrosis is important in predicting the prognosis [10]. Thus, risk and protective factors may differ according to the MASLD stage.

Over the past several years, various medical devices employing artificial intelligence (AI) and deep learning technologies have gained widespread popularity. Among these, AI software for DR screening, approved by the Food and Drug Administration, has been increasingly used. The proficiency of AI in analyzing posterior polar fundus photographs has been demonstrated to be comparable to that of retina specialists evaluating wide-angle fundus photographs [11]. A previous study reported that the AI software RetCAD achieved a 95.1% area under the receiver operating characteristic (ROC) curve for detecting DR, although the severity was not determined, with a standard error of 90.1% and standard deviation of 90.6% [12].

This study aimed to establish a standardized method for evaluating DR using AI for the first time and explore the correlation between DR and each MASLD stage at a large health screening facility in Japan.

## Materials and methods

### Study setting and data sources

This cross-sectional study was conducted at a health screening center at Omiya City Clinic, located in Tokyo's metropolitan area. Specifically, 33,022 patients who underwent fundus examination and blood tests from April 2015 to March 2016 were included. This study was approved by the Ethics Committees of Jikei University School of Medicine (31–428[10010]), Tokai University (20R-005), and Omiya City Clinic (No. 20). All studies were conducted in compliance with the tenets of the Declaration of Helsinki. Information was disclosed to the

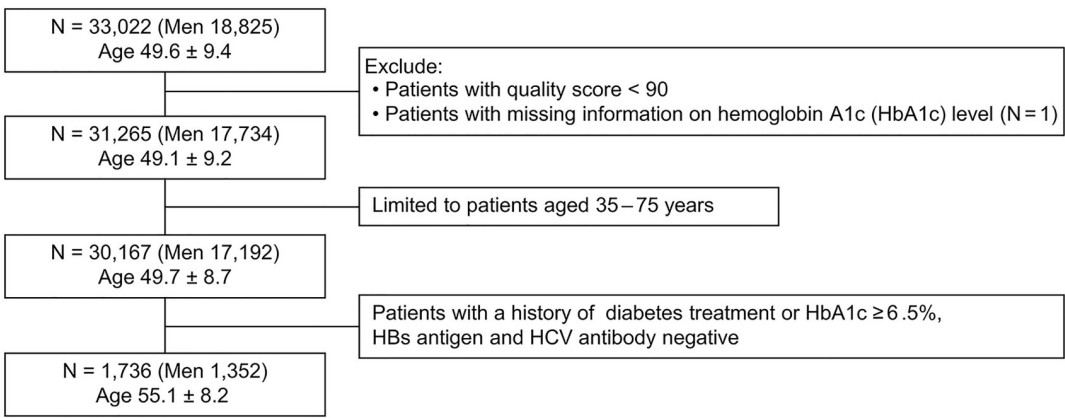

**Fig 1. Flow diagram.**

participants through an opt-out option on the Omiya City Clinic web page. Informed consent was obtained from all the participants by the opt-out method according to the Ethical Guidelines for Medical and Health Research Involving Human Subjects (Japanese Ministry of Health, Labour and Welfare). On April 8, 2022, data were accessed for research purposes.

## Study participants

For this cross-sectional study, we analyzed 33,022 records from both the eyes. Records with quality score (QS) < 90 were excluded, and records with the highest DR scores (DRSs) in either eye with the same ID were selected. A total of 30,167 participants aged 35–75 years were selected, of which 1,736 with a history of diabetes treatment or HbA1c level $\geq$ 6.5% and negative hepatitis B surface antigen and hepatitis C virus antibodies were included in the analysis (Fig 1).

## Data collection and measurement

## Quantification of fundus images and definition of cutoff values

Fundus photographs were captured using a non-mydriatic digital fundus camera (CR-2 PlusAF; Canon, Tokyo, Japan), focusing on the macula with a 45˚ field of view in both eyes. Considering the quality of fundus images at Omiya City Clinic and the compatibility with AI software, we used RetCAD (version 1.3.1; Thirona Bio, Inc., San Diego, CA, USA) for analysis. To ensure anonymity, the images were processed using Mity Safety Exporter before being uploading to RetCAD, where the DRSs were assessed on a scale of 0–100. A threshold for the QS of the fundus images, which was automatically generated using RetCAD, was established through validation. Dr. KK, a retinal specialist, reviewed 58 fundus images and categorized them based on the QS. The ROC curve analysis revealed an area under the curve (AUC) of 0.85. The optimal balance point between sensitivity and specificity was identified at a QS of 78.83, indicated by the red dot on the curve (Fig 2). To ensure selection of highly precise fundus images, we targeted for a minimal false-positive rate (marked with a yellow dot) at QS = 88 (Fig 2). Consequently, fundus images with a QS exceeding 90 were deemed suitable for evaluation [13].

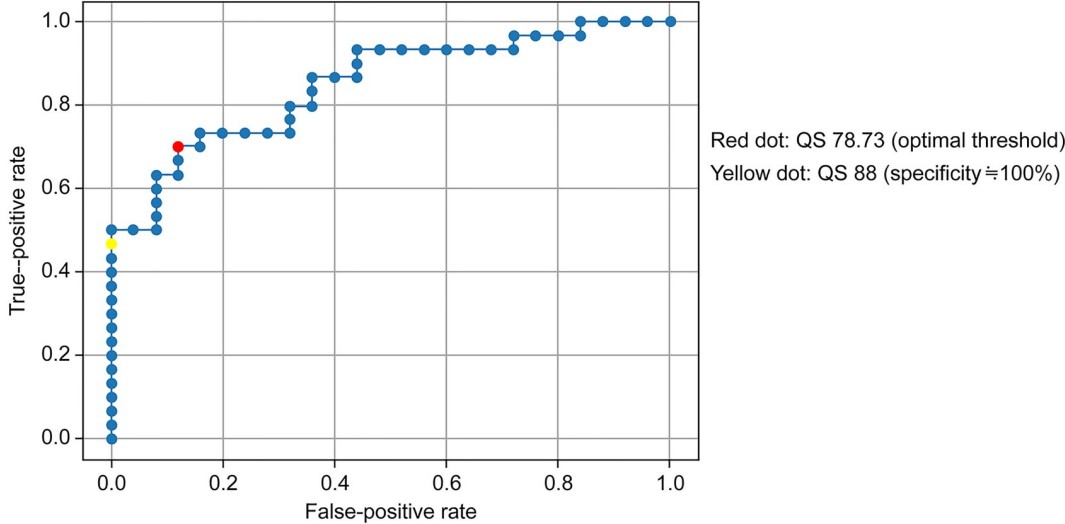

**Fig 2. Receiver operating characteristic curves for quality score cutoff value.**

## Definition of steatotic liver stage

Fatty liver stage was defined based on the presence or absence of steatotic liver (SL) on abdominal echocardiography and liver fibrosis indices, the Fibrosis-4 Index (FIB-4) and non-alcoholic fatty liver disease fibrosis score (NFS). Liver fibrosis indices were categorized into high and low groups based on the median value (FIB-4: 1.11, NFS: –1.03). The stages were classified as follows: SL0 (steatotic liver [–]), SL1 (steatotic liver (+)/low FIB-4), and SL2 (steatotic liver (+)/high FIB-4).

## Clinical parameters and lifestyle information

The following parameters were evaluated: body mass index (BMI); glycated hemoglobin (HbA1c); platelet count; and aspartate aminotransferase (AST), alanine aminotransferase (ALT), gamma-glutamyl transpeptidase (γ-GTP), cholinesterase (ChE), albumin, and C-reactive protein (CRP) levels. FIB-4, an index of liver fibrosis, was calculated using the following formula: FIB-4 = (age × AST)/(Plt count × square root of ALT). Similarly, the NFS was calculated with the following formula: NFS = -1.675 + 0.037 × age (years) + 0.094 × BMI (kg/m$^2$) + 1.13 × impaired fasting glycemia or diabetes (yes = 1, no = 0) + 0.99 × AST/ALT ratio—0.013 × platelet (×10$^9$ /L) − 0.66 × albumin (g/dL).

Alcohol consumption habits were obtained from Japan's questionnaire for general health examinations [14].

## Statistical analyses

Before validation, a cutoff value was established for the DRS. TheDr. KK evaluated 120 fundus images selected from the dataset and categorized them for the DRS using a binary scale as follows: 0 for no DR and 1 for suspected or probable DR. The cutoff values for the DRS were determined based on ROC curve analysis (Fig 3), which yielded an AUC of 0.92. The point with optimal balance between sensitivity and specificity, represented by a yellow dot, corresponded to a DRS of 39.24 (Fig 3). The point where sensitivity was maximized, near 100%, as indicated by the blue dot, was at a DRS of 23.14 (Fig 3). Conversely, the point where specificity

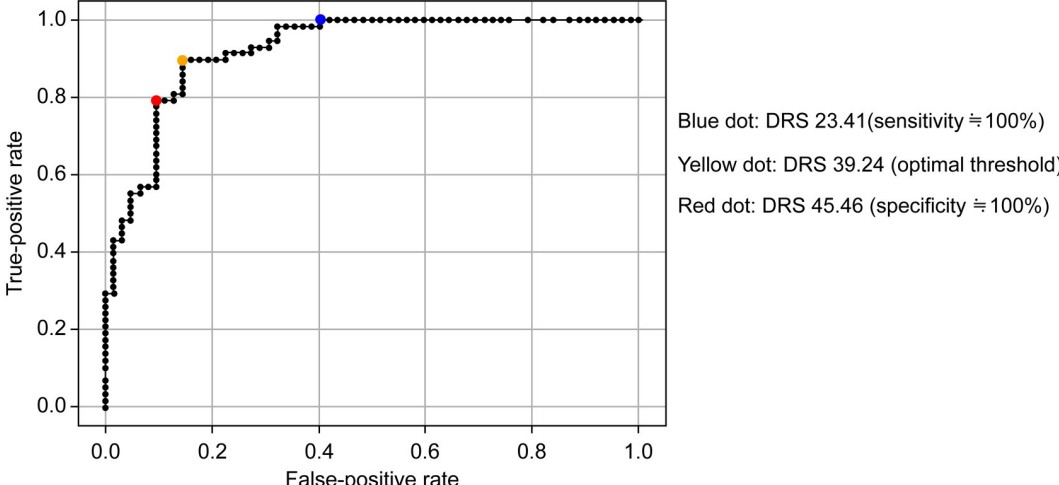

Blue dot: DRS 23.41(sensitivity ≒100%)

Yellow dot: DRS 39.24 (optimal threshold)

Red dot: DRS 45.46 (specificity ≒100%)

**Fig 3. Receiver operating characteristic curves for diabetic retinopathy score cutoff value.**

was maximized, approaching 100%, as indicated by the red dot, was at a DRS of 45.46 (Fig 3). Based on these findings, a DRS of 20 was selected as the threshold to differentiate between high (with DR) and low (no DR) DRS groups when aiming for approximately 100% sensitivity, whereas a DRS of 50 was selected for the same groups when targeting approximately 100% specificity.

Different factors were identified in the high-DRS and low-DRS groups in terms of MASLD. Logistic regression analysis was performed with cutoff values of 20 (sensitivity, 100%) and 50 (specificity, 100) [13]. Confounders were selected based on clinical importance (S1 Table), and included age, sex, BMI, HbA1c, CRP level, and alcohol consumption. SL stage was classified using both the FIB-4 and NFS.

## Results

The characteristics and major parameters for all participants are summarized in Table 1. The average age of participants was 55.1 ± 8.2 years, with a male predominance (77.9%). The average DRS was 17.7 ± 9.9 and average HbA1c level was 7.3 ± 1.2%. The proportions of SL stages based on the FIB-4 were 34.3% (n = 595) for SL0, 35.7% (n = 620) for SL1, and 30.0% (n = 521) for SL2. In contrast, the proportion of SL stages based on the NFS was 34.6% (n = 601) for SL1 and 31.1% (n = 540) for SL2, indicating that similar proportions were observed for both liver fibrosis indices.

Table 2 presents the participants' characteristics according to SL stage and DRS with a cutoff value of 20 (CO20). Using the FIB-4 classification, a high DRS was observed in 16.0% (n = 95) of SL0, 11.9% (n = 74) of SL1, and 11.3% (n = 59) of SL2 cases. The prevalence of high DRS decreased as the SL stage increased. Moreover, at all SL stages, more male participants than female participants had a high DRS.

Table 3 shows participants' characteristics according to SL stage and DRS with a cutoff value of 50 (CO50). For the FIB-4 classification, a high DRS was observed in 5.2% (n = 31) of SL0, 3.1% (n = 19) of SL1, and 2.9% (n = 15) of SL2 cases. Compared with CO20, a reduction was observed in the proportions owing to the higher cutoff standard, as well as a decreasing trend in the prevalence of a high DRS with increasing SL stage. Additionally, classifications based on the NFS showed similar tendencies for both CO20 and CO50 compared with those

**Table 1. Overall participant characteristics.**

| Continuous variables | |
|---|---|
| DRS | 17.7 ± 9.9 |
| HbA1c | 7.3 ± 1.2 |
| Age (years) | 55.1 ± 8.2 |
| ChE | 369.9 ± 69.3 |
| BMI | 26.1 ± 4.6 |
| CRP | 0.2 ± 0.5 |
| FIB-4 | 1.2 ± 0.6 |
| NFS | -1.1 ± 1.0 |
| Categorical variables | |
| Men, N (%) | 1352 (77.9) |
| Drinking (4 or more drinks), N (%) | 1145 (66.2) |
| SL stage 0, N (%) | 595 (34.3) |
| SL stage 1 (FIB-4), N (%) | 620 (35.7) |
| SL stage 2 (FIB-4), N (%) | 521 (30.0) |
| SL stage 1 (NFS), N (%) | 601 (34.6) |
| SL stage 2 (NFS), N (%) | 540 (31.1) |

Abbreviations: DRS, diabetic retinopathy score, HbA1c, glycated hemoglobin, ChE, cholinesterase, BMI, body mass index, CRP, C-reactive protein, FIB-4, Fibrosis-4 Index, NFS, non-alcoholic fatty liver disease fibrosis score
Continuous variables are shown as mean ± standard deviation.

**Table 2. Participant characteristics based on SL stages (FIB-4 and NFS) (cutoff = 20).**

| DRS cutoff = 20 | | SL0 | | SL1 (FIB-4) | | SL2 (FIB-4) | | SL1 (NFS) | | SL2 (NFS) | |
|---|---|---|---|---|---|---|---|---|---|---|---|
| | | DRS < 20 | DRS ≥ 20 | DRS < 20 | DRS ≥ 20 | DRS < 20 | DRS ≥ 20 | DRS < 20 | DRS ≥ 20 | DRS < 20 | DRS ≥ 20 |
| All | n (%) | 500 (84.0) | 95 (16.0) | 546 (88.1) | 74 (11.9) | 462 (88.7) | 59 (11.3) | 540 (89.9) | 61 (10.2) | 468 (86.7) | 72 (13.3) |
| | DRS | 14.7 ± 1.6 | 39.8 ± 17.0 | 14.3 ± 1.3 | 36.9 ± 16.2 | 14.7 ± 1.5 | 36.2 ± 14.8 | 14.3 ± 1.2 | 37.7 ± 16.9 | 14.7 ± 1.5 | 35.7 ± 14.5 |
| | Age | 57.3 ± 8.1 | 54.9 ± 8.6 | 51.0 ± 7.2 | 48.5 ± 6.6 | 58.2 ± 6.9 | 59.9 ± 7.0 | 51.2 ± 7.2 | 48.7 ± 6.9 | 57.9 ± 7.1 | 57.7 ± 8.2 |
| | HbA1c | 7.0 ± 1.1 | 7.5 ± 1.4 | 7.4 ± 1.2 | 8.2 ± 1.7 | 7.2 ± 1.1 | 7.6 ± 1.3 | 7.4 ± 1.2 | 8.1 ± 1.7 | 7.2 ± 1.1 | 7.8 ± 1.4 |
| | ChE | 345.9 ± 66.1 | 342.4 ± 70.2 | 395.6 ± 65.3 | 377.5 ± 64.7 | 372.9 ± 67.6 | 346.7 ± 55.5 | 399.1 ± 66.8 | 380.3 ± 63.5 | 369.2 ± 64.2 | 349.9 ± 58.5 |
| Men | n (%) | 376 (82.6) | 79 (17.4) | 409 (87.0) | 61 (13.0) | 377 (88.3) | 50 (11.7) | 405 (89.0) | 50 (11.0) | 381 (86.2) | 61 (13.8) |
| | DRS | 14.8 ± 1.6 | 39.9 ± 16.4 | 14.4 ± 1.3 | 37.9 ± 17.0 | 14.7 ± 1.5 | 37.4 ± 15.3 | 14.3 ± 1.2 | 39.2 ± 17.4 | 14.8 ± 1.6 | 36.4 ± 15.1 |
| | Age | 57.0 ± 7.9 | 54.5 ± 8.5 | 50.7 ± 7.2 | 48.6 ± 6.5 | 58.1 ± 7.1 | 59.6 ± 7.0 | 50.9 ± 7.2 | 48.9 ± 6.9 | 57.8 ± 7.3 | 57.4 ± 8.1 |
| | HbA1c | 7.0 ± 1.1 | 7.4 ± 1.4 | 7.4 ± 1.2 | 8.1 ± 1.6 | 7.2 ± 1.0 | 7.7 ± 1.3 | 7.3 ± 1.1 | 8.1 ± 1.7 | 7.2 ± 1.1 | 7.8 ± 1.3 |
| | ChE | 344.8 ± 65.4 | 339.7 ± 62.9 | 394.0 ± 65.3 | 370.2 ± 63.6 | 369.3 ± 66.7 | 344.0 ± 55.0 | 396.4 ± 66.3 | 374.4 ± 61.8 | 367.0 ± 64.6 | 345.3 ± 57.7 |
| Women | n (%) | 124 (88.6) | 16 (11.4) | 137 (91.3) | 13 (8.7) | 85 (90.4) | 9 (9.6) | 135 (92.5) | 11 (7.5) | 87 (88.8) | 11 (11.2) |
| | DRS | 14.5 ± 1.3 | 39.3 ± 20.4 | 14.3 ± 1.3 | 32.5 ± 11.9 | 14.6 ± 1.5 | 30.0 ± 10.2 | 14.4 ± 1.3 | 31.0 ± 13.0 | 14.5 ± 1.5 | 31.8 ± 9.3 |
| | Age | 57.9 ± 8.6 | 56.5 ± 9.1 | 51.7 ± 7.1 | 48.2 ± 7.5 | 58.7 ± 6.1 | 61.7 ± 7.3 | 52.0 ± 7.3 | 47.7 ± 7.3 | 58.0 ± 6.5 | 59.6 ± 8.7 |
| | HbA1c | 6.9 ± 1.1 | 7.6 ± 1.5 | 7.4 ± 1.2 | 8.5 ± 2.1 | 7.3 ± 1.3 | 7.3 ± 1.1 | 7.5 ± 1.3 | 8.2 ± 1.9 | 7.2 ± 1.1 | 7.8 ± 1.7 |
| | ChE | 349.4 ± 68.5 | 355.9 ± 100.4 | 400.4 ± 65.4 | 411.5 ± 61.0 | 388.9 ± 69.8 | 361.4 ± 59.3 | 407.2 ± 68.0 | 406.7 ± 67.7 | 378.5 ± 62.2 | 375.4 ± 59.0 |

Abbreviations: DM, diabetes mellitus, DRS, diabetic retinopathy score, SL, steatotic liver, HbA1c, glycated hemoglobin, FIB-4, Fibrosis-4 Index, NFS, non-alcoholic fatty liver disease fibrosis score, SL0, no SL, SL1, SL with low fibrosis, SL2, SL with high fibrosis, ChE, cholinesterase
Continuous variables are shown as mean ± standard deviation.

**Table 3. Participant characteristics based on SL stages (FIB-4 and NFS) (cutoff = 50).**

| DRS cutoff = 50 | | SL0 | | SL1 (FIB-4) | | SL2 (FIB-4) | | SL1 (NFS) | | SL2 (NFS) | |
|---|---|---|---|---|---|---|---|---|---|---|---|
| | | DRS < 50 | DRS ≥ 50 | DRS < 50 | DRS ≥ 50 | DRS < 50 | DRS ≥ 50 | DRS < 50 | DRS ≥ 50 | DRS < 50 | DRS ≥ 50 |
| **All** | **n (%)** | 564 (94.8) | 31 (5.2) | 601 (96.9) | 19 (3.1) | 506 (97.1) | 15 (2.9) | 583 (97.0) | 18 (3.0) | 524 (97.0) | 16 (3.0) |
| | **DRS** | 16.4 ± 5.5 | 61.3 ± 8.5 | 15.6 ± 4.9 | 61.1 ± 7.5 | 15.9 ± 5.0 | 57.3 ± 7.2 | 15.3 ± 4.3 | 60.9 ± 7.5 | 16.3 ± 5.5 | 57.8 ± 7.4 |
| | **Age** | 57.1 ± 8.2 | 53.1 ± 7.8 | 50.8 ± 7.1 | 47.1 ± 6.2 | 58.4 ± 6.9 | 58.9 ± 8.0 | 51.0 ± 7.2 | 48.4 ± 7.2 | 57.9 ± 7.2 | 56.6 ± 9.3 |
| | **HbA1c** | 7.0 ± 1.2 | 7.7 ± 1.6 | 7.4 ± 1.2 | 9.5 ± 1.6 | 7.2 ± 1.1 | 8.2 ± 1.5 | 7.4 ± 1.2 | 9.3 ± 1.7 | 7.2 ± 1.1 | 8.5 ± 1.5 |
| | **ChE** | 346.3 ± 66.8 | 328.1 ± 63.8 | 393.8 ± 64.8 | 380.8 ± 85.3 | 370.1 ± 67.3 | 366.2 ± 49.7 | 397.5 ± 66.3 | 388.3 ± 81.0 | 366.8 ± 64.0 | 358.7 ± 56.9 |
| **Men** | **n (%)** | 429 (94.3) | 26 (5.7) | 452 (96.2) | 18 (3.8) | 413 (96.7) | 14 (3.3) | 438 (96.3) | 17 (3.7) | 427 (96.6) | 15 (3.4) |
| | **DRS** | 16.7 ± 5.9 | 60.4 ± 7.7 | 15.7 ± 4.8 | 61.2 ± 7.7 | 16.0 ± 5.1 | 57.8 ± 7.2 | 15.3 ± 4.2 | 61.0 ± 7.7 | 16.3 ± 5.5 | 58.3 ± 7.4 |
| | **Age** | 56.8 ± 8.1 | 52.7 ± 7.4 | 50.6 ± 7.1 | 47.5 ± 6.1 | 58.2 ± 7.1 | 58.8 ± 8.3 | 50.8 ± 7.2 | 48.9 ± 7.1 | 57.8 ± 7.3 | 56.4 ± 9.6 |
| | **HbA1c** | 7.1 ± 1.2 | 7.4 ± 1.5 | 7.4 ± 1.2 | 9.3 ± 1.6 | 7.2 ± 1.1 | 8.2 ± 1.5 | 7.3 ± 1.2 | 9.1 ± 1.7 | 7.2 ± 1.1 | 8.5 ± 1.5 |
| | **ChE** | 345.1 ± 65.4 | 324.2 ± 54.2 | 391.1 ± 64.7 | 384.9 ± 85.8 | 366.1 ± 66.6 | 373.7 ± 41.8 | 394.0 ± 65.6 | 393.1 ± 80.7 | 364.0 ± 64.5 | 365.2 ± 52.3 |
| **Women** | **n (%)** | 135 (96.4) | 5 (3.6) | 149 (99.3) | 1 (0.7) | 93 (98.9) | 1 (1.1) | 145 (99.3) | 1 (0.7) | 97 (99.0) | 1 (1.0) |
| | **DRS** | 15.6 ± 4.2 | 65.9 ± 12.1 | 15.6 ± 5.2 | 58.7 ± 0.0 | 15.7 ± 4.3 | 50.5 ± 0.0 | 15.3 ± 4.5 | 58.7 ± 0.0 | 16.1 ± 5.4 | 50.5 ± 0.0 |
| | **Age** | 57.9 ± 8.6 | 54.8 ± 10.5 | 51.4 ± 7.2 | 40.0 ± 0.0 | 59.0 ± 6.3 | 60.0 ± 0.0 | 51.8 ± 7.3 | 40.0 ± 0.0 | 58.2 ± 6.8 | 60.0 ± 0.0 |
| | **HbA1c** | 6.9 ± 1.1 | 8.7 ± 1.9 | 7.5 ± 1.3 | 11.9 ± 0.0 | 7.3 ± 1.3 | 7.5 ± 0.0 | 7.5 ± 1.4 | 11.9 ± 0.0 | 7.2 ± 1.2 | 7.5 ± 0.0 |
| | **ChE** | 350.2 ± 71.4 | 348.6 ± 107.6 | 402.0 ± 64.6 | 306.0 ± 0.0 | 387.6 ± 68.2 | 261.0 ± 0.0 | 407.9 ± 67.5 | 306.0 ± 0.0 | 379.4 ± 60.7 | 261.0 ± 0.0 |

Abbreviations: DM, diabetes mellitus, DRS, diabetic retinopathy score, SL, steatotic liver, HbA1c, glycated hemoglobin, FIB-4, Fibrosis-4 Index, NFS, non-alcoholic fatty liver disease fibrosis score, SL0, no SL, SL1, SL with low fibrosis, SL2, SL with high fibrosis, ChE, cholinesterase

Continuous variables are shown as mean ± standard deviation.

based on the FIB-4. Regarding ChE, those with SL1 and SL2, indicating the presence of SL, displayed higher ChE levels than did those with SL0. Conversely, across all SL stages, participants with a high DRS consistently showed lower ChE levels than did those with a low DRS.

Table 4 presents the results of logistic regression analysis conducted to identify risk factors for a high DRS using CO20 and CO50. Variables were stratified by SL stage and adjusted for HbA1c levels. Confounding factors (age, sex, BMI, HbA1c, γ-GTP, CRP, and alcohol consumption) were selected based on clinical significance. Using CO20, the adjusted odds ratio (OR) (95% confidence interval) for the risk of DR in the S1 and S2 groups was evaluated using the S0 group as the reference. For the FIB-4, the values for the S1 and S2 groups were 0.63 (0.43–0.92) and 0.64 (0.43–0.95), respectively. Similarly, for the NFS, the values were 0.54 (0.37–0.79) and 0.77 (0.52–1.13) for the S1 and S2 groups, respectively. A negative correlation was observed between MASLD and DR. Moreover, as the liver fibrosis index increased, the risk of DR increased slightly. Further analysis by dividing ChE levels into tertiles showed that in the high ChE level group ($> 399$), adjusted OR for the risk of DR was 0.52 (0.34–0.79) for the FIB-4 and 0.54 (0.36–0.83) for the NFS. As the ChE levels increased, the risk of DR decreased ($p < 0.01$). Similarly, these trends were corroborated for CO50 in the FIB-4 and NFS.

## Discussion

In this study, we established a standardized method for evaluating DR using AI and explored its correlation with MASLD across various stages. The findings revealed that while SL disease itself is associated with a reduced risk of developing DR, the risk of DR slightly increases with higher liver fibrosis indices. Additionally, ChE levels were identified as an independent factor contributing to the reduced risk of DR. To the best of our knowledge, this study is the first

**Table 4. Risk factors for a high DRS stratified by SL stages and adjusted for HbA1c (multivariate analysis).**

| DRS cutoff = 20 | | SL stages stratified by FIB-4 | | | SL stages stratified by NFS | | |
|---|---|---|---|---|---|---|---|
| | | OR | 95% CI | | OR | 95% CI | |
| Age | | 0.99 | 0.97 | 1.01 | 0.98 | 0.96 | 1.003 |
| Sex (men) | | 1.54 | 1.04 | 2.28 | 1.52 | 1.03 | 2.25 |
| SL | stage 0 | 1 (ref) | | | 1 (ref) | | |
| | stage 1 | 0.63 | 0.43 | 0.92 | 0.54 | 0.37 | 0.79 |
| | stage 2 | 0.64 | 0.43 | 0.95 | 0.77 | 0.52 | 1.13 |
| | p for trend | 0.02 | | | 0.13 | | |
| ChE | < 338 | 1 (ref) | | | 1 (ref) | | |
| | 338–399 | 0.89 | 0.59 | 1.34 | 0.90 | 0.60 | 1.35 |
| | > 399 | 0.52 | 0.34 | 0.79 | 0.54 | 0.36 | 0.83 |
| | p for trend | <0.01 | | | < 0.01 | | |
| BMI | | 1.02 | 0.99 | 1.06 | 1.02 | 0.98 | 1.05 |
| HbA1c | | 1.38 | 1.25 | 1.53 | 1.38 | 1.24 | 1.52 |
| CRP | | 0.93 | 0.69 | 1.25 | 0.93 | 0.69 | 1.25 |
| Drinking (4 or more drinks) | | 0.85 | 0.62 | 1.17 | 0.84 | 0.62 | 1.16 |
| **DRS cutoff = 50** | | **SL stages stratified by FIB-4** | | | **SL stages stratified by NFS** | | |
| | | OR | 95% CI | | OR | 95% CI | |
| Age | | 0.97 | 0.94 | 1.01 | 0.97 | 0.94 | 1.01 |
| Sex (men) | | 2.53 | 1.11 | 5.77 | 2.53 | 1.11 | 5.78 |
| SL | stage 0 | 1 (ref) | | | 1 (ref) | | |
| | stage 1 | 0.48 | 0.24 | 0.94 | 0.50 | 0.26 | 0.98 |
| | stage 2 | 0.57 | 0.28 | 1.15 | 0.53 | 0.25 | 1.10 |
| | p for trend | 0.09 | | | 0.07 | | |
| ChE | < 338 | 1 (ref) | | | 1 (ref) | | |
| | 338–399 | 0.83 | 0.41 | 1.67 | 0.83 | 0.41 | 1.66 |
| | > 399 | 0.41 | 0.20 | 0.85 | 0.40 | 0.19 | 0.84 |
| | p for trend | < 0.01 | | | < 0.01 | | |
| BMI | | 1.02 | 0.95 | 1.08 | 1.02 | 0.95 | 1.08 |
| HbA1c | | 1.58 | 1.36 | 1.83 | 1.58 | 1.36 | 1.83 |
| CRP | | 0.75 | 0.33 | 1.69 | 0.75 | 0.33 | 1.68 |
| Drinking (4 or more drinks) | | 0.88 | 0.51 | 1.53 | 0.89 | 0.51 | 1.53 |

Abbreviations: DRS, diabetic retinopathy score, HbA1c, glycated hemoglobin, OR, odds ratio, CI, confidence interval,

ChE, cholinesterase, BMI, body mass index, CRP, C-reactive protein, SL, steatotic liver, FIB-4, Fibrosis-4 Index, NFS, non-alcoholic fatty liver disease fibrosis score, ref, reference,

P-value for trend is calculated by converting SL stage or ChE level into continuous variables.

attempt to standardize the reading accuracy of fundus images using AI and clarify the relationship between DR and MASLD.

Determining cutoff values for AI-based judgments can be challenging. In this study, we established two DRS cutoff values (CO20 and CO50) that achieved 100% sensitivity and specificity, respectively. This approach enabled thorough risk exploration with 100% sensitivity, although it also included lesions beyond DR with 100% sensitivity. To address this, we used a 100% specificity threshold to confirm whether the observed results were specific to DR. Additionally, we combined abdominal ultrasound findings with two liver fibrosis indices to examine the relationship between DR and MASLD progression stages. Our results, derived from data at a Japanese health screening facility, suggest that individuals with SL disease may have a

lower risk of developing DR. Furthermore, ChE appears to contribute independently to the reduced risk of DR.

According to a meta-analysis by Zhang et al., no overall association was observed between DR and MASLD; however, the results varied depending on the type of diabetes [7]. Specifically, no association was found with type 2 DM, while a positive association was found with type 1 DM [7]. Further sub-analysis revealed a positive correlation among Caucasians and a negative correlation among Asians [7].

Several studies based on Japanese data support this trend [15]. Specifically, Hashimoto et al. [16], Tanabe et al. [17], and Takeuchi et al. [18] reported a negative correlation between DR and MASLD, whereas Yoneda et al. [19] confirmed a positive correlation between DR and MASLD in type 1 DM (S2 Table). In this study, we found a negative correlation between DR and MASLD. Based on data from Japanese health screening facilities, it can be inferred that type 2 DM is predominant [20]. This finding is consistent with the results of domestic reports [16–18], and a similar trend has been observed in Asian countries. The underlying reasons could be racial differences in insulin secretion capacity and resistance [21] and variations in dietary habits. Moreover, ChE was found to be an independent protective factor against DR. ChE is mainly classified as acetylcholinesterase or butyrylcholinesterase (BuChE). Notably, ChE in health screening indicates BuChE, which is synthesized in the liver and used as a marker for liver function. Patients with diabetes and SL reportedly have increased ChE values [22]. In our study, participants with steatotic liver in the SL1 and SL2 groups had higher ChE values than did those in the SL0 group, suggesting that steatotic liver might increase BuChE levels, potentially reducing the risk of DR. Furthermore, BuChE activity is associated with retinal blood flow through the blood–retinal barrier, and its activity reportedly decreases by 30–50% in the retinas of diabetes-induced mice models [23]. Therefore, BuChE may independently reduce the risk of DR. However, several aspects of this mechanism remain unclear, thus requiring further research.

Using abdominal ultrasound and the FIB-4 to determine the progression of MASLD, the presence of SL has been shown to reduce the risk of DR, whereas an increase in the FIB-4 increases the risk of DR. A positive correlation between liver fibrosis and DR has been reported [7]. This may be due to the expression of tumor necrosis factor-α by adipose tissue, promoting liver fibrosis and potentially causing retinal damage [24–26]. Furthermore, activation of the renin-angiotensin system and production of reactive oxygen species due to liver fibrosis may also be involved in DR progression [27, 28]. Recent research has shown that a potent nicotinamide adenine dinucleotide phosphate oxidase 4 inhibitor is effective in treating the initial pathological events of DR [29].

This study has several limitations. First, being a cross-sectional study, it is challenging to establish causality, and further longitudinal investigations are needed. Second, the effects of the duration and stage of DM could not be assessed. Thus, large-scale validation of this method is necessary. Third, our use of posterior-pole fundus images for DR evaluation might have led to the omission of peripheral retinal changes. The introduction of a wide-angle fundus imaging device or an AI software update could address this issue. Finally, the pre-validation assessment of the study population may have been insufficient. Future research should consider the most effective approach for pre-validation.

## Conclusions

Although numerous meta-analyses have reported an association between DR and MASLD, the findings remain inconsistent. In this study, we explored the relationship between DR and MASLD using high-level evaluation techniques, including standard AI-based analysis of

fundus images and SL stages. The results suggest that MASLD and ChE may act as protective factors against DR. Further research into the protective mechanisms and the standardization of DR evaluation is necessary.

## Supporting information

**S1 Table. Risk factors for high DRS (univariate analysis).**
(DOCX)

**S2 Table. Comparison of risk factors for DR and MASLD in Japanese studies.**
(DOCX)

## Acknowledgments

We thank all patients for their participation in this study.

## Author Contributions

**Conceptualization:** Koji Komatsu, Tadashi Nakano.

**Data curation:** Koji Komatsu, Kei Sano, Kota Fukai, Ryo Nakagawa, Takashi Nakagawa, Masayuki Tatemichi.

**Formal analysis:** Koji Komatsu, Kei Sano, Kota Fukai, Masayuki Tatemichi.

**Investigation:** Koji Komatsu.

**Methodology:** Koji Komatsu, Ryo Nakagawa, Takashi Nakagawa, Masayuki Tatemichi.

**Project administration:** Koji Komatsu, Tadashi Nakano.

**Resources:** Ryo Nakagawa, Takashi Nakagawa, Tadashi Nakano.

**Software:** Koji Komatsu, Tadashi Nakano.

**Supervision:** Masayuki Tatemichi, Tadashi Nakano.

**Validation:** Koji Komatsu, Masayuki Tatemichi.

**Visualization:** Koji Komatsu, Kei Sano.

**Writing – original draft:** Koji Komatsu, Kei Sano, Kota Fukai, Masayuki Tatemichi.

**Writing – review & editing:** Koji Komatsu, Kota Fukai, Masayuki Tatemichi.

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
