## [Decision Letter · Decision Letter 0]

4 Nov 2024

PONE-D-24-35486Standardized evaluation of diabetic retinopathy using artificial intelligence and its association with metabolic dysfunction-associated steatotic liver disease in Japan: A cross-sectional studyPLOS ONE

Dear Dr. komatsu,

Thank you for submitting your manuscript to PLOS ONE. After careful consideration, we feel that it has merit but does not fully meet PLOS ONE’s publication criteria as it currently stands. Therefore, we invite you to submit a revised version of the manuscript that addresses the points raised during the review process.

We look forward to receiving your revised manuscript.

Kind regards,

Anna Di Sessa, PhD, MD

Academic Editor

PLOS ONE

Journal Requirements:

2. In the online submission form, you indicated that the datasets generated and/or analyzed during the current study are not publicly available due to privacy protection and confidentiality agreements but are available from the corresponding author on reasonable request. 

Additional Editor Comments :

The study is interesting and well-organized. However, before considering the paper for potential publication, all the reviewers' comments need to be adequately addressed.

Reviewers' comments:

Reviewer's Responses to Questions

**Comments to the Author**

1. Is the manuscript technically sound, and do the data support the conclusions?

Reviewer #1: Yes

Reviewer #2: Yes

2. Has the statistical analysis been performed appropriately and rigorously? 

Reviewer #1: Yes

Reviewer #2: Yes

3. Have the authors made all data underlying the findings in their manuscript fully available?

Reviewer #1: No

Reviewer #2: Yes

4. Is the manuscript presented in an intelligible fashion and written in standard English?

Reviewer #1: Yes

Reviewer #2: Yes

5. Review Comments to the Author

Reviewer #1: The manuscript presents a technically sound study that explores the relationship between diabetic retinopathy and metabolic dysfunction-associated steatotic liver disease using artificial intelligence. The statistical analysis is well-executed, with appropriate adjustments for confounders like age and BMI. However, the data used in the study is not publicly available due to privacy concerns, which limits transparency. The authors could improve the manuscript by considering ways to share anonymized data. Additionally, the study uses an opt-out consent method, and it would be helpful to clarify how this aligns with ethical standards for protecting patient privacy.

Reviewer #2: Overall, the article is very well written and easily understandable. The theme itself is of relevance, given the prevalence of DM in the world population, and also it's growth, being DR one of the most damaging diabetes complications . On the medical and statistical aspects of this article, I have only one remark to make. In line 179, it's stated that " the prevalence of high DSR decreased as the SL stage increased, and this trend was consistent across sexes " when commenting about the reports show in table 2. But in said table, when you look and the numbers about the female population, that`s not the case. According to table 2, when using FIB4 classification, 8,7% of women staged as SL1 has DRS > 20 in opposition to 9, 6% of those staged as SL2. So, in this case, the trend observed among male`s in table 2 does not repeat itself in the female population. So, it`s unclear what the author`s meant with this statement. When it comes to the AI aspect of the presented work, there`s a fundamental question that should have been answered : why did the author`s chose RetCAD for this work? The only reference to this question that I`ve found in the text is in the lines 73-76 : "A previous study reported that the AI software RetCAD achieved a 95.1% area under the receiver operating characteristic (ROC) curve for detecting DR, although the severity was not determined, with a standard error of 90.1% and standard deviation of 90.6%". But this information does not answer the question, because a good tool to solve a problem is not, necessarily, the best tool for such problem. There`s plenty of research about AI analysis of retinal imaging, with a myriad of softwares available for testing and validation. There`s also different forms of analysis, as recognized in the text in the statement in lines 281- 283 " Third, our use of posterior-pole fundus images for DR evaluation might have led to the omission of peripheral retinal changes. The introduction of a wide-angle fundus imaging device or an AI software update could address this issue". Why did the author`s chose posterior-pole fundus image for this particular work them? Was it budgetary constraints? Software limitations? What`s the reason for utilizing only fundus imaging, instead of a multi-modal AI system that incorporates other forms of data such as angiography and optic coherence tomography? For more information about multi-modal systems in ophthalmology , see https://doi.org/10.1186/s40662-024-00405-1. The reasoning behind the choice of the AI system used in the work is important, because the author`s claim to have " established a standardized method for evaluating DR using AI and explored its correlation with MASLD across various stages" in lines 230-231. If other`s, reading this article, decide to validate and improve in the method`s proposed, it is important for them to known the motive`s for software choice. Was it because of it`s accuracy? Cost? Commercial availability? Widespread adoption? If the criteria utilized for such a choice was cost-benefit, for example, researchers working in a more high risk environment could consider chosing a different software, with better accuracy and higher cost. Because of the reasons previously stated, I sugest to be included in the text the motives behind the adoption of RetCAD for this work.

6. PLOS authors have the option to publish the peer review history of their article (what does this mean?). If published, this will include your full peer review and any attached files.

Reviewer #1: No

Reviewer #2: No

---

## [Author Response · Author response to Decision Letter 0]

28 Nov 2024

Emily Chenette Editor-in-Chief

Anna Di Sessa Academic Editor

It is a pleasure to communicate with you again in relation to the peer revision of our manuscript. First of all, I would like to thank the reviewers with whose comments the value of this paper increased immensely. Please note the following changes made in the manuscript in accordance with the reviewers’ comments. 

Additionally, the text added based on the advice is expressed in red, while the original text is in italics.

We sincerely hope that the current version of the manuscript is acceptable for publication in the PLOS ONE.

Koji Komatsu

Reviewer #1

♯1. The manuscript presents a technically sound study that explores the relationship between diabetic retinopathy and metabolic dysfunction-associated steatotic liver disease using artificial intelligence. The statistical analysis is well-executed, with appropriate adjustments for confounders like age and BMI. However, the data used in the study is not publicly available due to privacy concerns, which limits transparency. The authors could improve the manuscript by considering ways to share anonymized data. Additionally, the study uses an opt-out consent method, and it would be helpful to clarify how this aligns with ethical standards for protecting patient privacy.

Response:

Thank you for your feedback. This study does not involve any interventions, human biological specimens, or personally identifiable information requiring special consideration. Therefore, it complies with the "Ethical Guidelines for Medical and Health Research Involving Human Subjects" issued by the Ministry of Health, Labour, and Welfare. Based on your feedback, we have added more detailed content to the Materials and Methods section.

Revisions：From line 87 to line 92.

All studies were conducted in compliance with the tenets of the Declaration of Helsinki. Information was disclosed to the participants through an opt-out option on the Omiya City Clinic web page. Informed consent was obtained from all the participants by the opt-out method according to the Ethical Guidelines for Medical and Health Research Involving Human Subjects (Japanese Ministry of Health, Labour and Welfare). On April 8, 2022, data were accessed for research purposes.

Furthermore, due to ethical regulations, the analyzed large dataset has not been made publicly available. However, it can be provided upon request for legitimate purposes. The content regarding this information has also been revised.

Revisions：From line 300 to line 304.

Data Availability Statement

Data cannot be shared publicly because of ethical restrictions. Data are available from the Institutional Review Board of the Jikei University School of Medicine for researchers who meet the criteria for access to confidential data. The Jikei University Hospital Ethics Committee Secretariat 3-25-8 Nishi-Shimbashi, Minato-ku Tokyo, Japan, 105-8461 TEL:+81-3-3433-1111.

Reviewer #2

♯1. In line 179, it's stated that " the prevalence of high DSR decreased as the SL stage increased, and this trend was consistent across sexes " when commenting about the reports show in table 2. But in said table, when you look and the numbers about the female population, that`s not the case. According to table 2, when using FIB4 classification, 8,7% of women staged as SL1 has DRS > 20 in opposition to 9, 6% of those staged as SL2. So, in this case, the trend observed among male`s in table 2 does not repeat itself in the female population. So, it`s unclear what the author`s meant with this statement. 

Response:

Thank you for your thorough review. As you pointed out, there was a notation error, which has been corrected as shown below.

Revisions：From line 179 to line 183.

Table 2 presents the participants’ characteristics according to SL stage and DRS with a cutoff value of 20 (CO20). Using the FIB-4 classification, a high DRS was observed in 16.0% (n = 95) of SL0, 11.9% (n = 74) of SL1, and 11.3% (n = 59) of SL2 cases. The prevalence of high DRS decreased as the SL stage increased. Moreover, at all SL stages, more male participants than female participants had a high DRS.

♯2. 

When it comes to the AI aspect of the presented work, there`s a fundamental question that should have been answered : why did the author`s chose RetCAD for this work? The only reference to this question that I`ve found in the text is in the lines 73-76 : "A previous study reported that the AI software RetCAD achieved a 95.1% area under the receiver operating characteristic (ROC) curve for detecting DR, although the severity was not determined, with a standard error of 90.1% and standard deviation of 90.6%". But this information does not answer the question, because a good tool to solve a problem is not, necessarily, the best tool for such problem. There`s plenty of research about AI analysis of retinal imaging, with a myriad of softwares available for testing and validation. There`s also different forms of analysis, as recognized in the text in the statement in lines 281- 283 " Third, our use of posterior-pole fundus images for DR evaluation might have led to the omission of peripheral retinal changes. The introduction of a wide-angle fundus imaging device or an AI software update could address this issue". Why did the author`s chose posterior-pole fundus image for this particular work them? Was it budgetary constraints? Software limitations? What`s the reason for utilizing only fundus imaging, instead of a multi-modal AI system that incorporates other forms of data such as angiography and optic coherence tomography? For more information about multi-modal systems in ophthalmology , see https://doi.org/10.1186/s40662-024-00405-1. The reasoning behind the choice of the AI system used in the work is important, because the author`s claim to have " established a standardized method for evaluating DR using AI and explored its correlation with MASLD across various stages" in lines 230-231. If other`s, reading this article, decide to validate and improve in the method`s proposed, it is important for them to known the motive`s for software choice. Was it because of it`s accuracy? Cost? Commercial availability? Widespread adoption? If the criteria utilized for such a choice was cost-benefit, for example, researchers working in a more high risk environment could consider chosing a different software, with better accuracy and higher cost. Because of the reasons previously stated, I sugest to be included in the text the motives behind the adoption of RetCAD for this work.

Response:

Thank you for your valuable comments. This study was conducted at Omiya City Clinic, a health screening facility in Japan. Japan has a unique preventive healthcare practice known as "Ningen Dock." In ophthalmologic examinations for "Ningen Dock," it is common to conduct visual acuity tests, intraocular pressure measurements, and posterior fundus photography assessments. This practice remained unchanged both in 2015, when data collection for this study was conducted, and as of 2024. Although optical coherence tomography (OCT) and wide-angle fundus photography have become widespread in clinics and hospitals, they are still not commonly introduced in health screening facilities. Therefore, the evaluation in this study was based on posterior fundus photography.

The major advantage of conducting research at health screening facilities is the ability to collect large-scale data from a diverse age group. When handling large-scale data, image diagnostics can be prone to error, so we implemented AI-based standardized evaluations.

When this study was designed in 2020, only a few software programs for AI-based fundus image evaluation were available for use in Japan. Considering the quality of fundus images at Omiya City Clinic and compatibility with AI software, we selected RetCAD from Throna, a startup at that time.

We also believe that introducing multi-dimensional systems, such as OCT and wide-angle fundus photography, could lead to even more interesting results in handling a large number of subjects efficiently, and we look forward to future research in this area.

Based on the above, I have added the reasons for selecting RetCAD to the Materials and methods section."

Revisions：From line 105 to line 110.

Fundus photographs were captured using a non-mydriatic digital fundus camera (CR-2 PlusAF; Canon, Tokyo, Japan), focusing on the macula with a 45° field of view in both eyes. Considering the quality of fundus images at Omiya City Clinic and the compatibility with AI software, we used RetCAD (version 1.3.1; Thirona Bio, Inc., San Diego, CA, USA) for analysis. To ensure anonymity, the images were processed using Mity Safety Exporter before being uploaded to RetCAD, where the DRSs were assessed on a scale of 0–100.

We thank you for the time and effort spent throughout the review process of this manuscript. We believe that the excellent comments provided by all did increase the utility of this paper. We hope we were able to revise the paper in accordance with the reviewers’ expectations. We are honored to be working on this paper with you throughout this revision process.

---

## [Editor Report · Decision Letter 1]

1 Dec 2024

Standardized evaluation of diabetic retinopathy using artificial intelligence and its association with metabolic dysfunction-associated steatotic liver disease in Japan: A cross-sectional study

PONE-D-24-35486R1

Dear Dr. komatsu,

We’re pleased to inform you that your manuscript has been judged scientifically suitable for publication and will be formally accepted for publication once it meets all outstanding technical requirements.

Kind regards,

Anna Di Sessa, PhD, MD

Academic Editor

PLOS ONE
---

## [Editor Report · Acceptance letter]

5 Dec 2024

PONE-D-24-35486R1 

PLOS ONE

Dear Dr. Komatsu, 

I'm pleased to inform you that your manuscript has been deemed suitable for publication in PLOS ONE. Congratulations! Your manuscript is now being handed over to our production team.

Kind regards, 

on behalf of

Dr. Anna Di Sessa 

Academic Editor

PLOS ONE